# Comparison of the Prognostic Value of Ki-67 and Programmed Cell Death Ligand-1 in Patients with Upper Tract Urothelial Carcinoma

**DOI:** 10.3390/jcm10163728

**Published:** 2021-08-21

**Authors:** Mu-Yao Tsai, Ping-Chia Chiang, Chien-Hsu Chen, Ming-Tse Sung, Shun-Chen Huang, Jau-Ling Suen, Eing-Mei Tsai, Po-Hui Chiang

**Affiliations:** 1Department of Urology, Kaohsiung Chang Gung Memorial Hospital, Chang Gung University College of Medicine, Kaohsiung 83301, Taiwan; kingtsaiking@gmail.com (M.-Y.T.); pingagaga@hotmail.com (P.-C.C.); kenkochen@yahoo.com.tw (C.-H.C.); 2Graduate Institute of Medicine, College of Medicine, Kaohsiung Medical University, Kaohsiung 80708, Taiwan; jlsuen@kmu.edu.tw; 3Division of Natural Science, College of Liberal Education, Shu-Te University, Kaohsiung 82445, Taiwan; 4Department of Anatomic Pathology, Kaohsiung Chang Gung Memorial Hospitale, Chang Gung Univerity College of Medicine, Kaohsiung 83301, Taiwan; mtsmts@cgmh.org.tw (M.-T.S.); shuang@cgmh.org.tw (S.-C.H.); 5College of Medicine, Kaohsiung Medical University, Kaohsiung 80708, Taiwan; 6Research Center for Environmental Medicine, Kaohsiung Medical University, Kaohsiung 80708, Taiwan; 7Department of Obstetrics and Gynecology, Kaohsiung Medical University Hospital, Kaohsiung Medical University, Kaohsiung 80708, Taiwan

**Keywords:** combined positive score, Ki-67, programmed cell death ligand-1, radical nephroureterectomy, upper tract urothelial carcinoma

## Abstract

We retrospectively enrolled 102 patients with upper tract urothelial carcinoma (UTUC) who underwent radical nephroureterectomy to examine the prognostic value of Ki-67 and programmed cell death ligand-1 (PD-L1). Then, we performed PD-L1 and Ki-67 immunohistochemical staining on whole tissue sections. The cut-off value of PD-L1 positivity was a combined positive score (CPS) ≥10 and the Ki-67 overexpression was 20%. Among the 102 patients, 16.7% and 48.0% showed positive PD-L1 expression and Ki-67 overexpression, respectively. A CPS ≥10 was significantly associated with a higher pathological T stage (*p* = 0.049). In addition, Ki-67 overexpression was significantly associated with a pathological T stage ≥ 2 (*p* = 0.027) and tumour necrosis (*p* = 0.016). In the multivariable analysis, a positive PD-L1 expression was significantly correlated with worse cancer-specific survival (HR = 3.66, 95% CI =1.37−9.77, *p* = 0.01). However, there was no predictive value using a combination of PD-L1 expression and Ki-67 overexpression as a prognostic predictor. Compared with Ki-67 overexpression, a positive PD-L1 expression with CPS ≥ 10 was a stronger independent prognostic factor for CSS in patients with UTUC.

## 1. Introduction

Upper tract urothelial carcinoma (UTUC) accounts for only around 5% of all urothelial carcinoma (UC) cases [1]. However, in Taiwan, the prevalence of UTUC is unusually high, accounting for more than 10% of all UC cases; this high prevalence of UTUC may be attributed to the high prevalence of chronic kidney disease [1,2,3,4,5]. In addition, arsenic-contaminated well water has been associated with a high prevalence of UC [1,6,7,8]. Moreover, the prevalence of UTUC was found to be higher in herbalists [9]. Although UTUC has been indicated to have poor prognosis and oncological outcomes, accurate prognostic factors for UTUC are unavailable. Many studies have attempted to develop a prognostic model based on postoperative pathological or preoperative parameters; however, novel biomarkers and predictive prognostic models remain under debate [10,11].

Biomarkers may be crucial in predicting a UTUC prognosis. Programmed cell death-1 positivity was found to have an independent prognostic value of cancer-specific survival (CSS) outcomes in patients with UTUC receiving extirpative surgery [12]. Moreover, high-risk clinicopathological factors for UTUC were associated with programmed cell death ligand-1 (PD-L1) positivity [13]. Ki-67, a nuclear cell proliferation marker that can be examined through immunostaining, was frequently found to be expressed in malignant cancers such as breast, colon, and ovarian cancers. A positive Ki-67 expression has been associated with adverse pathological features and poor recurrence- and cancer-free survival in patients with UTUC receiving nephroureterectomy [14,15,16,17,18]. Among the various biomarkers examined previously, Ki-67 appears to be the most potential biomarker with its cell proliferation characteristic [10,11,19]. In our previous study, we found that a PD-L1 combined positive score (CPS) ≥ 10 in UTUC was associated with worse CSS and overall survival (OS) [20]. However, few studies have evaluated the prognostic value of Ki-67 and PD-L1 in combination. In the present study, we investigated the prognostic value of Ki-67 and PD-L1 in UTUC.

## 2. Materials and Methods

### 2.1. Study Population

Between 2013 and 2018, a total of 102 patients with UTUC who underwent radical nephroureterectomy (RNU) through laparoscopic or traditional methods were retrospectively enrolled in this study. We excluded patients without PD-L1 or Ki-67 immunochemical stains, bladder cancer stage T2 or higher, and patients with distant metastasis at diagnosis or receiving neoadjuvant chemotherapy. Information regarding patients’ specimens was obtained from the Surgical Pathology Database of Kaohsiung Chang Gung Memorial Hospital (KCGMH). This study was approved by the Institutional Review Board of KCGMH (Approval No: 202000185B0).

Lymph node staging was performed by radiologists or the treating physician based on the computed tomography (CT) scan findings. Lymph node dissection (LND) was performed during the operation when enlarged lymph nodes were noted. The pathological tumour stage was determined according to the eighth edition of the American Joint Committee on Cancer’s staging system for cancer of the renal pelvis and ureter.

Locoregional recurrence was defined as the presence of a recurrent lesion at the previous operation site or the evidence of retroperitoneal lymph node enlargement on the CT scan during follow-up. Distant metastasis was defined as the evidence of suspicious lesions on radiological images. In addition, recurrent bladder cancer was determined on the basis of pathological reports. The history of bladder cancer and the cause of death were determined through a chart review.

### 2.2. Immunohistochemistry

Specimen staining was performed by a specialised pathologist (SMZ). The tissue samples submitted for the PD-L1 study were fixed in formalin, embedded in paraffin, and sectioned into 3-μm thickness. The immunohistochemical examination of PD-L1 was performed using a Dako anti-PD-L1 (clone 22C3) mouse monoclonal primary antibody purchased from Agilent Technologies (Santa Clara, CA, USA). The detection system used was the EnVision FLEX Visualisation System (Dako North America, Inc., Carpinteria, CA, USA). Immunostaining was performed using a Dako Autostainer Link 48 (Dako North America, Inc., Carpinteria, CA, USA). Figure 1 shows a representative picture of positive and negative PD-L1 expressions observed through immunohistochemical staining.

The specimens for Ki-67 immunostaining were fixed in 10% formalin, embedded in paraffin, and sectioned into 1-μm thickness. The tissue sections were deparaffinised using xylene and a graded ethanol series and then heated in boiling 0.01-M citrate buffer (pH 6.0) for 15 min in a microwave oven for antigen retrieval. Subsequently, the sections were incubated with the primary mouse monoclonal anti-Ki-67 antibody (clone MIB-1; Dako, Glostrup, Denmark) diluted in the antibody diluent reagent solution (ScyTek, Logan, UT, USA) at room temperature for 60 min, rinsed with a buffer, amplified using the Quanto Amplifier and Quanto HRP polymer (Thermo Scientific, Kalamazoo, MI, USA), labelled with DAB chromogen (Bio SB, Santa Barbara, CA, USA), and then counterstained with haematoxylin. A negative reagent control was set, in which the patient tissue was processed using the same steps, except for incubation with the primary antibody. A small section of the tonsil tissue attached on each experimental slide was used as the positive (germinal centre) and negative (mature squamous epithelium) tissue control.

### 2.3. Immunohistochemical Analysis

The combined positive score (CPS) was used to examine the PD-L1 expression in the tissue sections. The CPS was defined as the ratio of the number of PD-L1-stained cells (tumour cells, lymphocytes, and macrophages) to the total number of tumour cells. We performed a time-dependent receiver operating characteristic (ROC) curve and survival ROC curve by the SAS system to find the cut-off value of PD-L1 in this study. A CPS ≥ 10 was found to be the optimal cut-off value of this study CPS of <10 and ≥10 indicated negative and positive PD-L1 expressions, respectively (Figure 1).

We visually calculated the Ki-67 index by counting the total number of stained tumour cells and the total number of tumour cells in each high-power field for a total of five high-power fields. The Ki-67 index was calculated as the number of stained tumour cells divided by the total number of tumour cells and then multiplied by 100. The cut-off value of Ki-67 overexpression was also calculated by the SAS system by the same methods. Ki-67 overexpression was considered when the samples exhibited a nuclear reactivity of ≥20% (Figure 1).

### 2.4. Statistical Analysis

The relationships among the PD-L1 expression, Ki-67 overexpression, clinical characteristics, and pathological features were examined using the chi-square test or Fisher’s exact test. The Kaplan–Meier method was used to evaluate the survival function, and the differences were examined using the log-rank test. Univariable and multivariable Cox proportional hazard regression models by forward stepwise selection for the OS, as well as CSS, were used to assess the prognostic indicators—namely, sex, age at diagnosis, PD-L1 expression, Ki-67 overexpression, tumour node metastasis classification, and other clinicopathological characteristics. The level of significance was set at *p* < 0.05. All statistical analyses were performed using SPSS (IBM Corp. Released 2011. IBM SPSS Statistics for Windows, Version 20.0. Armonk, NY: IBM Corp. USA).

## 3. Results

### 3.1. Patient Characteristics

Table 1 lists the patients’ demographic characteristics. The median age of all 102 patients was 69 years, and 62 patients were aged >65 years. Of the 102 patients, 51 (50.0%) were men and 51 (50.0%) were women; moreover, 40 (39.2%) were diagnosed as having pathological T3 or higher UTUC, and 14 (13.7%) exhibited lymph nodes of >10 mm in size on their CT findings at diagnosis. In addition, six (5.9%), eight (7.8%), and eight (7.8%) patients received palliative immunotherapy (pembrolizumab was the main regimen), adjuvant chemotherapy, and palliative chemotherapy, respectively.

### 3.2. Association of PD-L1 Expression and Ki-67 Overexpression with Clinicopathological Characteristics

Of the 102 patients, 18 and 49 exhibited positive PD-L1 expression and Ki-67 overexpression, respectively. The results are presented in Table 2. The follow-up duration was significantly shorter in patients with a CPS of ≥10 (19.8 ± 20.5 vs. 37.2 ± 24.0 months, *p* = 0.006) and those with a Ki-67 index of ≥20 (28.4 ± 20.9 vs. 39.9 ± 25.9 months, *p* = 0.016).

A PD-L1-positive expression was associated with a higher T stage (*p* = 0.049), higher proportion of patients receiving palliative immunotherapy (23.5% vs. 2.4%, *p* = 0.007), and lower proportion of patients exhibiting bladder tumour recurrence after an operation during follow-up (0% vs. 28.2%, *p* = 0.01). A higher proportion of patients with a pathological T stage ≥ 2 showed Ki-67 overexpression (71.4% vs. 49%, *p* = 0.027). Furthermore, the patients with Ki-67 overexpression showed a higher proportion of tumour necrosis in specimens (45.1% vs. 30.2%, *p* = 0.016).

### 3.3. Association of Evaluated Predictors and Survival Outcomes

#### 3.3.1. Kaplan–Meier Analysis

The Kaplan–Meier analysis results showed that a positive PD-L1 expression was associated with a shorter CSS (*p* < 0.002; Figure 2). A Ki-67 overexpression was not associated with the OS (*p* = 0.082) or CSS (*p* = 0.368; Figure 2).

#### 3.3.2. Multivariable Analysis for OS and CSS

In the multivariable analysis using forward stepwise selection for the OS (Table 3), age over 65 years (HR = 2.96, 95% CI = 1.55–5.65, *p* < 0.001) and a higher T stage were also found to be independent risk factors for a shorter OS. In addition, a positive PD-L1 expression did not show a significant correlation with the OS, and the combination of CPS ≥ 10 and Ki-67 ≥ 20% did not show significance. Furthermore, a positive PD-L1 expression (HR = 3.66, 95% CI = 1.37–9.77, *p* = 0.01), a lymph node ≥10 mm in size on a CT scan (HR = 5.06, 95% CI = 1.97–13.02, *p* = 0.001), receiving palliative chemotherapy (HR = 2.80, 95% CI = 1.02–7.64, *p* = 0.045), and a higher T stage were independent risk factors for shorter CSS. The combination of positive PD-L1 and overexpression of Ki-67 was not correlated with CSS after the analysis.

## 4. Discussion

According to the 2020 European Association of Urology (EAU) guidelines, several factors are associated with a poor prognosis of UTUC, including a high pathological stage, older age, multifocality, and hydronephrosis [21]. These guidelines were based on the findings of some studies examining the prognostic value of biomarkers intraoperatively and postoperatively. In a study conducted by Skala et al., they reported that PD-L1 positivity was associated with a high histological grade, a high pathological stage, and angiolymphatic invasion [13]. Another study conducted in 2017 reported that PD-L1 expression in UTUC cells independently predicted a shorter CSS [22]. Miyama et al. performed a retrospective study by recruiting 271 patients with UTUC who underwent nephroureterectomy and found that a high platelet count and PD-L1 positivity were significantly associated with a shorter metastasis-free survival [23]. However, most of these studies did not find an association between PD-L1 positivity and CSS or OS. Moreover, the aforementioned studies were limited by their small sample sizes and retrospective design. In this study, we defined PD-L1 positivity by CPS (a combination of tumour cells and immune cells). Several studies found that it was more helpful to incorporate tumour-associated immune cells into the positivity of PD-L1 to select responders than using tumour cells alone [24,25]. Besides, it is easily and repeatable to calculate the tumour cells and immune cells at the same time. Therefore, CPS may be a feasible and better way to predict the prognosis.

A prospective study was conducted by Krabbe, L.M. et al. to investigate the prognostic value of Ki-67 in patients with UTUC who underwent RNU [14]. This study indicated an association between Ki-67 overexpression and adverse pathological features. In addition, poor CSS and recurrence-free survival were associated with Ki-67 overexpression. However, the percentage of Ki-67 overexpression in urothelial carcinoma in their study was higher than previous research (73.3% vs. 42.5–50%) [26,27]. As a result, the interpretation of the outcomes might be affected by the high percentage of Ki-67 positivity in their study. There were 48% patients presenting Ki-67 overexpression in our study, and this was compatible with the percentage of Ki-67 ≥ 20% in previous studies [26,27]. This might explain the different results between these studies. In another study, Krabbe et al. retrospectively examined 475 patients and reported that Ki-67 overexpression was an independent predictor of the CSS and recurrence-free survival of patients with UTUC who underwent RNU [15]. Since there were discrepancies in the results in each model and a limitation of the retrospective analysis in most studies, this suggests that it is more reasonable to combine these known predictive markers from various pathways into a multivariable prognostic model than to only examine the single markers.

In our study, we found that Ki-67 overexpression was not associated with CSS and OS in the multivariable analysis. Although we enrolled Asian patients in our study, our results differed from those reported for Japanese and Korean patients. A cohort study examined and compared the pathological characteristics and behaviours of patients with UTUC between Taiwan and Japan [7]. The study reported that a higher proportion of Japanese patients exhibited lymphovascular invasion, whereas a higher proportion of Taiwanese patients demonstrated squamous differentiation. This unusual etiological difference between Japanese and Taiwanese patients can be because the Taiwanese patients with UTUC had different genetic susceptibilities or carcinogen exposure and tended to be non-smokers. The non-association of Ki-67 overexpression with CSS and OS observed in our study may partly be attributed to racial differences.

In this study, we investigated the Ki-67 overexpression and PD-L1 expression simultaneously. A shorter follow-up duration was found both in patients with PD-L1-positive expression and Ki-67 overexpression, which may be attributed to the poorer prognosis in these patients. Additionally, a CPS ≥ 10 was associated with a higher pathological T stage. Ki-67 overexpression was only associated with pathological T2 or higher. However, the linear-by-linear association test showed that Ki-67 overexpression was associated with a higher T stage (*p* = 0.030,). Therefore, these results also implied that PD-L1 positivity may have a stronger association with a worse prognosis than Ki-67 overexpression.

Few studies have examined the predictive value of Ki-67 expression in combination with PD-L1 positivity in patients with UTUC receiving RNU. To our knowledge, only one study, conducted by Rubino et al. in 2020, demonstrated that positive Ki-67 and PD-L1 expression in patients with muscle-invasive bladder cancer receiving neoadjuvant chemotherapy were associated with the OS [28]. They also found that the pathological N stage was associated with a poor OS [28]. To our knowledge, this is the first study to examine the prognostic value of Ki-67 overexpression and PD-L1 positivity at the same time in patients with UTUC receiving nephroureterectomy. No synergic effect was observed in the multivariable analysis when we combined Ki-67 overexpression and positive PD-L1 expression as a risk factor. PD-L1 positivity showed a stronger prognostic value than Ki-67 overexpression for CSS.

Our study was limited by its retrospective small-scale study and relatively shorter follow-up period (34.17 months). The prevalence of upper tract urothelial carcinoma is relatively low in comparison with bladder cancer. To minimise the specimen heterogeneity in this study, we only enrolled patients meeting the selection criteria, and this may have further minimised our sample size. Second, the lymph node status of most patients was determined on the basis of images. In addition, we did not routinely perform LND during RNU, because the benefits of LND are still under debate. We performed LND only if suspicious enlarged lymph nodes (>10 mm) were observed on the CT images before RNU or palpable enlarged lymph nodes were observed during the operation. Therefore, some nodal metastasis bias might have occurred. Third, given the relatively small sample size and few events, we failed to include all the established prognostic factors of OS or CSS in the multivariable analysis. However, we combined the prognostic variables with the most impact, such as immunochemical staining and the T and N stages, to evaluate the OS and CSS. Due to its higher prognostic value for CSS, PD-L1 may serve as a potential marker in treatment-related decision-making. The predictive value of PD-L1 should be examined in further prospective large-scale studies.

## 5. Conclusions

The results indicated that a positive PD-L1 expression with CPS ≥ 10 was significantly associated with the pathological T stage. Additionally, a Ki-67 overexpression of ≥20% demonstrated a significant association with pathological T ≥ 2 and tumour necrosis. Compared with Ki-67 overexpression, positive PD-L1 expression was found to be a stronger independent prognostic factor for CSS in patients with UTUC.

## Figures and Tables

**Figure 1 jcm-10-03728-f001:**
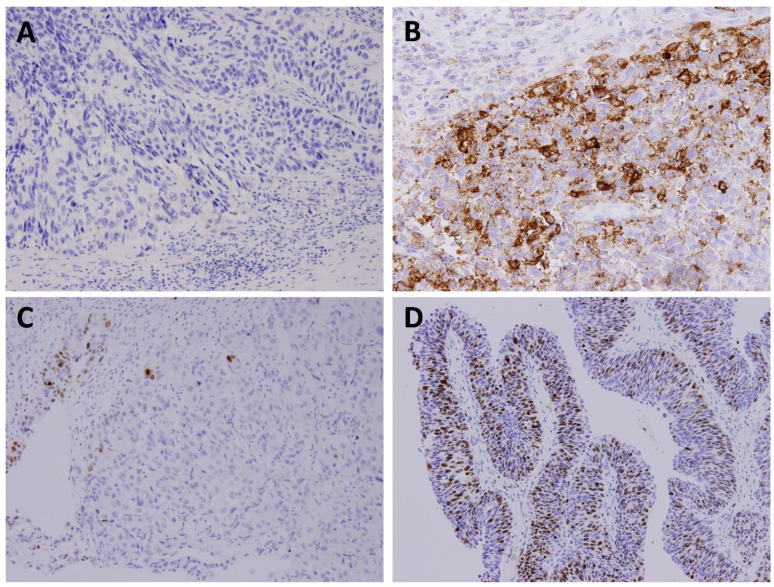
Representative photomicrographs showing PD-L1 expression through immunostaining (200×): (**A**) Negative PD-L1 expression (CPS < 10) and (**B**) PD-L1 expression (CPS = 50). Representative photomicrographs showing Ki-67 expression: (**C**) 5% and (**D**) 70%.

**Figure 2 jcm-10-03728-f002:**
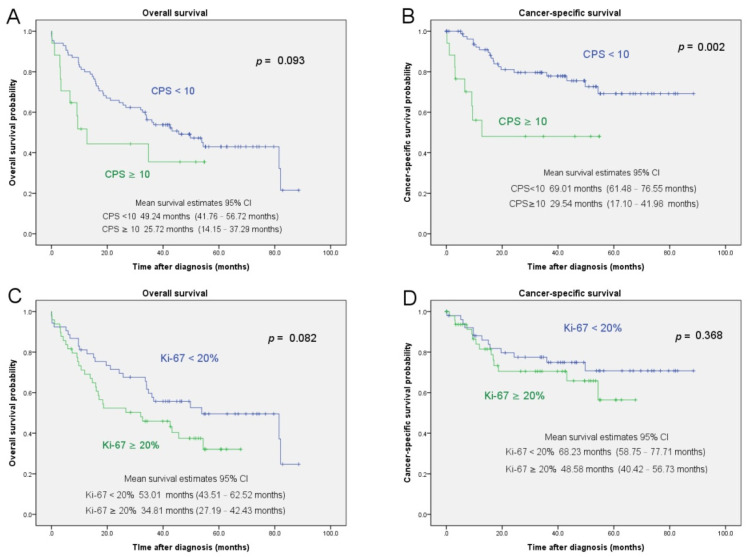
The OS and CSS estimates after RNU for patients with UTUC based on PD-L1 expression and Ki-67 overexpression. Subfigure (**A**,**B**) showed probability of OS and CSS after RNU stratified according to PD-L1 expression, respectively. Subfigure (**C**,**D**) showed probability of OS and CSS after RNU stratified according to Ki-67 expression, respectively.

**Table 1 jcm-10-03728-t001:** Demographic characteristics of the 102 patients with UTUC.

Variables	*N* (%)
Age >65 years	62 (60.8)
Gender	
Male	51 (50.0)
Female	51 (50.0)
Lymph node >10 mm on CT at diagnosis	
Absent	88 (86.3)
Present	14 (13.7)
Pathologic stage	
pTa/is	25 (24.5)
pT1	16 (15.7)
pT2	21 (20.6)
pT3	32 (31.4)
pT4	8 (7.8)
Adjuvant chemotherapy	8 (7.8)
Palliative immunotherapy	6 (5.9)
Palliative chemotherapy	8 (7.8)
PD-L1 CPS ≥ 10	17 (16.7)
Ki-67 ≥ 20	49 (48.0)

**Table 2 jcm-10-03728-t002:** The association of PD-L1 expression and Ki-67 expression with the clinicopathological characteristics in patients with UTUC.

	PD-L1 CPS ≥ 10	PD-L1 CPS < 10	*p*-Value	Ki-67 ≥ 20	Ki-67 < 20	*p*-Value
	*N* (%)	*N* (%)		*N* (%)	*N* (%)	
Patient number	17 (16.7)	85 (83.3)		49(48.0)	53(52.0)	
Follow duration	19.8 ± 20.5 (mo)	37.2 ± 24.0 (mo)	0.006 *	28.4 ± 20.9 (mo)	39.9 ± 25.9 (mo)	0.016 *
Age			0.468			0.192
>65 yr	9 (52.9)	53 (62.4)		33 (67.3)	29 (54.7)	
≤65 yr	8 (47.1)	32 (37.6)		16 (32.7)	24 (45.3)	
Gender			0.184			1.000
Male	11 (64.7)	40 (47.1)		24 (49.0)	27 (50.9)	
Female	6 (35.3)	45 (52.9)		25 (51.0)	26 (49.1)	
Pathologic stage			0.049 *			0.160
pTa/is	0 (0)	25 (29.4)		7 (14.3)	18 (34.0)	
pT1	3 (17.6)	13 (15.3)		7 (14.3)	9 (17.0)	
pT2	5 (29.4)	16 (18.8)		12 (24.4)	9 (17.0)	
pT3	7 (41.2)	25 (29.4)		19 (38.8)	13 (24.5)	
pT4	2 (11.8)	6 (7.1)		4 (8.2)	4 (7.5)	
Muscle-invasive tumour			0.056			0.027 *
<T2	3 (17.6)	38 (44.7)		14 (28.6)	27 (50.9)	
≥T2	14 (82.4)	47 (55.3)		35 (71.4)	26 (49.1)	
Lymph node status			0.055			0.395
N0/Nx	12 (70.6)	76 (89.4)		44 (89.8)	44 (83.0)	
N1/N2	5 (29.4)	9 (10.6)		5 (10.2)	9 (17.0)	
Papillary feature			0.142			0.133
Absent	8 (47.1)	22 (25.9)		18 (36.7)	12 (22.6)	
Present	9 (52.9)	63 (74.1)		31 (63.3)	41 (77.4)	
Tumour grade			0.255			0.054
Low	0 (0)	11 (12.9)		2 (4.1)	9 (17.0)	
High	17 (100)	74 (87.1)		47 (95.9)	44 (83.0)	
Lymphovascular invasion			0.156			0.093
Absent	9 (52.9)	60 (70.6)		29 (59.2)	40 (75.5)	
Present	8 (47.1)	25 (29.4)		20 (40.8)	13 (24.5)	
Concomitant CIS			1.000			0.690
Absent	7 (41.2)	35 (41.2)		19 (38.8)	23 (43.4)	
Present	10 (58.8)	50 (58.8)		30 (61.2)	30 (56.6)	
Squamous differentiation			0.363			0.825
Absent	11 (61.1)	65 (76.5)		36 (73.5)	40 (75.5)	
Present	6 (38.9)	20 (23.5)		13 (26.5)	13 (24.5)	
Margin positive			1.000			1.000
Negative	15 (88.2)	75 (88.2)		43 (87.8)	47 (88.7)	
Positive	2 (11.8)	10 (11.8)		6 (12.2)	6 (11.3)	
Tumour necrosis			0.179			0.016 *
Absent	7 (41.2)	52 (61.2)		22 (44.9)	37 (69.8)	
Present	10 (58.8)	33 (38.8)		27 (45.1)	16 (30.2)	
Multifocal tumour			0.772			0.521
Solitary	13 (76.5)	59 (69.4)		33 (67.3)	39 (73.6)	
Multifocal	4 (23.5)	26 (30.6)		16 (32.7)	14 (26.4)	
Bladder recurrence			0.010 *			0.495
Yes	0 (0)	24 (28.2)		10 (20.4)	14 (26.4)	
No	17 (100)	61 (71.8)		39 (79.6)	39 (73.6)	
Locoregional recurrence			0.222			0.641
Yes	6 (38.9)	18 (21.2)		13 (26.5)	11 (20.8)	
No	11 (61.1)	67 (78.8)		36 (73.5)	42 (79.2)	
Distant metastasis			0.518			1.000
Yes	5 (29.4)	17 (18.8)		11 (22.4)	11 (20.8)	
No	12 (70.6)	68 (81.2)		38 (77.6)	42 (79.2)	
Adjuvant chemotherapy			1.000			1.000
Yes	1 (5.9)	7 (8.2)		4 (8.2)	4 (7.5)	
No	16 (94.1)	78 (91.8)		45 (71.8)	49 (92.5)	
Palliative chemotherapy			0.617			1.000
Yes	2 (11.8)	6 (7.1)		4 (8.2)	4 (7.5)	
No	15 (88.2)	79 (92.9)		45 (71.8)	49 (92.5)	
Palliative immunotherapy			0.007 *			0.102
Yes	4 (23.5)	2 (2.4)		5 (10.2)	1 (1.9)	
No	13 (76.5)	83 (97.6)		44 (89.8)	52 (98.1)	
Death due to UTUC			0.067			0.661
Yes	8 (47.1)	19 (22.4)		14 (28.6)	13 (24.5)	
No	9 (52.9)	66 (77.6)		35 (71.4)	40 (75.5)	

CIS, carcinoma in situ; * indicates *p* < 0.05.

**Table 3 jcm-10-03728-t003:** Univariable and multivariable analyses by the forward stepwise selection of the clinicopathological features for the prediction of CSS and OS in patients with UTUC.

	Cancer-Specific Survival				Overall Survival			
	Univariable		Multivariable *		Univariable		Multivariable *	
	HR (95% CI)	*p*-Value	HR (95% CI)	*p*-Value	HR (95% CI)	*p*-Value	HR (95% CI)	*p*-Value
Age (year)								
>65 vs. ≤65	2.18 (0.95−5.03)	0.067			3.14 (1.68−5.88)	<0.001	2.96 (1.55−5.65)	<0.001 *
PD-L1 expression								
CPS ≥ 10 vs. <10	3.46 (1.51−7.96)	0.003	3.66 (1.37−9.77)	0.01 *	1.79 (0.90−3.57)	0.098		
Ki-67 overexpression								
≥20% vs. <20%	1.41 (0.66–3.01)	0.371			1.60 (0.94–2.72)	0.085		
PDL1 and Ki-67 expression								
Both positive vs. negative	3.90 (1.46–10.40)	0.006			2.07 (0.88–4.87)	0.094		
T stage		<0.001		0.021 *		0.001		0.001 *
Ta/is	Referent		Referent		Referent		Referent	
T1	1.64 (0.23−11.63)	0.622	0.86 (0.12−6.36)	0.881	1.85 (0.68−5.02)	0.226	1.61 (0.59−4.36)	0.354
T2	3.18 (0.58−17.42)	0.183	1.82 (0.30−11.07)	0.515	2.44 (0.99−5.99)	0.053	2.28 (0.93−5.61)	0.072
T3	6.97 (1.52−32.02)	0.013	3.22 (0.65−15.98)	0.152	3.76 (1.62−8.77)	0.002	2.79 (1.18−6.57)	0.019 *
T4	27.42 (5.70−131.87)	<0.001	8.96 (1.62−49.51)	0.012 *	8.10 (2.94−22.30)	<0.001	8.30 (2.98−23.12)	<0.001 *
Lymph node status								
N1/N2 vs. N0/Nx	6.84 (3.07−15.25)	<0.001	5.06 (1.97−13.02)	0.001 *	2.90 (1.48−5.69)	0.002		
Adjuvant chemotherapy								
Yes vs. No	3.08 (1.14−8.30)	0.026			1.79 (0.76−4.23)	0.182		
Palliative chemotherapy								
Yes vs. No	4.53 (1.91−10.74)	0.001	2.80 (1.02–7.64)	0.045 *	1.96 (0.89−4.34)	0.097		
Palliative immunotherapy								
Yes vs. No	3.21 (0.95−10.86)	0.061			1.50 (0.47−4.86)	0.495		

* Variables significant in the univariable analysis were included to perform the multivariable analysis by forward stepwise selection.

## Data Availability

The data that support the findings of this study are available from the corresponding author upon reasonable request.

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
