# Peer review of "Comparison of the Prognostic Value of Ki-67 and Programmed Cell Death Ligand-1 in Patients with Upper Tract Urothelial Carcinoma"

_jcm, 2021, doi:10.3390/jcm10163728_

Round 1
Reviewer 1 Report
It was a pleasure reviewing the manuscript "Comparison of the prognostic value of Ki-67 and programmed cell death ligand-1 in patients with upper tract urothelial carcinoma". The authors report positive PD-L1 expression was significantly correlated with worse cancer-specific survival. However the gist of this manuscript has already been published (reference 20) by the authors. This manuscript is not bring anything new to the medical literature. Minor comment-- Fig. 2 needs to be high qualityAuthor Response
Reviewer #1:
It was a pleasure reviewing the manuscript "Comparison of the prognostic value of Ki-67 and programmed cell death ligand-1 in patients with upper tract urothelial carcinoma". The authors report positive PD-L1 expression was significantly correlated with worse cancer-specific survival. However the gist of this manuscript has already been published (reference 20) by the authors. This manuscript is not bring anything new to the medical literature. Minor comment-- Fig. 2 needs to be high quality
Reply:
We appreciate your positive review for our manuscript. Compared with previously published study, this study indicated longer follow-up duration. Also, PD-L1 CPS is only related to cancer-specific survival, not overall survival in this study. We hope that the updated result may help physicians to treat upper tract urothelial carcinoma. Besides, we have uploaded high resolution (DPI:600) figures(Figure 1 and 2) to improve the quality. We inserted the high resolution images as attached.

Reviewer 2 Report
The authors responded sufficiently to my points althoufgh I disagree that thickness of a slide does not affect number of ki67-positive cells.
Author Response
Reviewer #2:
The authors responded sufficiently to my points althoufgh I disagree that thickness of a slide does not affect number of ki67-positive cells.
Reply:
Thank you for your contributive suggestions. In our institute, we use microwave for antigen retrieval. During heating in microwave, the tissue is frequently fall off if we cut 3 µm thick. So we adjust to 1 µm thick. We have compared 3 µm and 1 µm thick section for Ki-67 counting on more than 30 cases of breast cancer. And the result count discrepancy is less than 3%, mainly because on 1 µm section some positive cells look weaker and may be interpreted as negative. In fact, for nuclear staining, 1 µm thick is more easily to count the cell number due to less overlapping.
(As prognostic markers of the breast, HER-2, ER, PR are done by fully automated machine which antigen retrieval is proceeded on heating plate that tissue fall off problem happens much less frequently. But of course the cost is much higher. This fully automated machine recommends 3 µm thick section. That's why we did the comparison -- just to make sure even Ki-67 not done by automated machine, the counting is still reliable.)

Round 2
Reviewer 1 Report
I again reviewed the manuscript "Comparison of the prognostic value of Ki-67 and programmed cell death ligand-1 in patients with upper tract urothelial carcinoma" and the manuscript "Prognostic value of PD-L1 combined positive score in patients with upper tract urothelial carcinoma" published in Cancer Immunol Immunother . 2021 March. Online ahead of print.
In my view this is duplicate publication. The results are not significantly different to justify publication.
This manuscript is a resubmission of an earlier submission. The following is a list of the peer review reports and author responses from that submission.
Round 1
Reviewer 1 Report
It was a pleasure reviewing the manuscript "Predicting the prognosis in upper tract urothelial carcinoma: identifying the optimal biomarker" by Tsai and colleagues. The manuscript is well written. I have some minor grammatical corrections only.
- In the abstract define OS and CSS.
- In abstract-- modify the statement "In multivariable analysis, positive PD-L1 expression was significantly correlated with OS" to state inferior or worse OS and same with CSS.
- Page 4 line 145 -- pembrolizumab is generic name and should not start with capital
- Line 157 should state higher death due to UTUC
- Figure 2 needs to be high quality
Reviewer 2 Report
The Authors tried to to examine the prognostic value of Ki-67 and programmed cell death ligand-1 (PD-L1) in the setting of UTUC.
They analysed data of 102 pts, in multivariable analysis, positive PD-L1 expression was significantly correlated with OS and CSS but there was no predictive value using combination of PD-L1 expression and Ki-67 overexpression as a prognostic predictor. Positive PD-L1 expression was a stronger independent prognostic factor Compared with Ki-67 over- expression.
The paper is well written, statistical analysis sounds good. Neverthelss, The role ki 67 is well known in the setting of urothelial cancer, the role of ki 67+ PDL1 had no predictive value in this study.
However the study suffers of several limits:
retrospective design
too short follow up (less than 2 yrs)
low number of involved patients
I suggest at least a longer follow up (by updating follow up of the patients) and a higher number of enrolled patients.
Reviewer 3 Report
In their manuscript entitled „Predicting the prognosis in upper tract urothelial carcinoma: identifying the optimal biomarker“ Tsai and coworkers assessed the value in terms of a prognostic biomarker for Ki67 and PD-L1 expression.
The manuscript is well written and easy to follow.
I have some points that should be addressed:
- Images are of poor quality and for PD-L1 expression with CPS of 20, a more informative image should be chosen. In Fig. 1b it looks like a CPS of 100. The same is true for Fig. 1c/d with Ki67-staining. Please use images that are more illustrative.
- Why are Ki67-immunostains fixed in 10% and not 4% formalin and cut at 1µm thickness while PD-L1-slides were cut at 3µm?
- The title is a bit misleading, as “only” two biomarkers were analyzed. Please re-phrase accordingly.